# A "Red-and-Green Porcelain" Figurine from a Jin Period Archaeological Site in the Primor'ye Region, Southern Russian Far East

**Irina S. Zhushchikhovskaya [1,\*] and Igor Yu Buravlev [2,3]**

1   Institute of History, Archaeology & Ethnology of Peoples of Far East, Far Eastern Branch of Academy of Sciences, 89, Pushkinskaya Str., 690001 Vladivostok, Russia
2   Institute of Chemistry, Far Eastern Branch of Academy of Sciences, 159, Prosp. 100-Letiya Vladivostoka, 690022 Vladivostok, Russia
3   Far Eastern Federal University, 10 Ajax Bay, Russky Island, 690922 Vladivostok, Russia
*   Correspondence: irina1zh@mail.ru

**Abstract:** This paper considers the results of an examination of a polychrome glazed anthropomorphic ceramic figurine from the Prmor'ye region (southern Russian Far East) discovered at one of the Jin period (1115–1234 CE) archaeological sites. The study attests to the hypothesis about the attribution of this unique art object to the "red-and-green porcelain" produced in Northern China since the mid-Jin period. At present "the red-and-green porcelain" is the object of certain research interest as an important stage of Chinese ceramics history preceding the invention of famous porcelains with overglazed enamel decoration. The main technological features and material properties of the studied object were determined using analytical methods of optical and electron (SEM) microscopy with the use of X-ray elemental composition analysis (EDS, pXRF). The main result of the study presented in the paper includes evidence that the polychrome ceramic figurine found at the Anan'evka walled town in Primor'ey in the south of the Russian Far East belongs to the category of "red-and-green porcelain", or "red-green ware". As supposed, the figurine portrays Zen monk Budai—a person popular in Chinese arts and spiritual culture of the Song and Jin periods. Therefore, the polychrome ceramics figurine from the Primor'ye region may be considered today as the most northeastern case of "red-and-green porcelain" discovered in an archaeological context.

**Keywords:** southern Russian Far East; Jin period (1115–1234 CE); archaeological remains; Northern China; ceramics; glazes; SEM-EDS; pXRF; optical microscopy

## 1. Introduction

In the Pimor'ye region of the southern Russian Far East neighboring northeast China many archeological sites of the Jin period, 1115–1234 CE, have been discovered and excavated since the end of the 1950s. The large territory of the modern Primor'ye region was part of the Jurchen Jin Empire between 1115 and 1218 CE, and then, from 1218 to 1233 CE was part of the separated Jurchen Dong Xia state. Among the most interesting and important categories of artefacts unearthed at the Jurchen sites in the Primor'ye region are porcelains and glazed ceramic wares. These are mostly table service items—various dishes, bowls, cups, bottles, etc. Ritual such as incenses and toilet accessories such as cosmetic boxes and flasks are present in small numbers [1]. Recent investigations based on physicochemical methods argue that such porcelains as Ding and their imitations, Jun, celadons, and some others were produced in the kilns of Northern China [2].

There is a single art object within the assemblage of porcelains and glazed ceramics from Jin period sites of the Primor'ye region. The fragments of a polychrome glazed ceramic figurine were found in 1977 at the Anan'evka walled settlement dated to XII–XIII c. (Figure 1). Based on some iconographic features the researchers interpreted this figurine as preliminarily a kind of bodhisattva image [3].

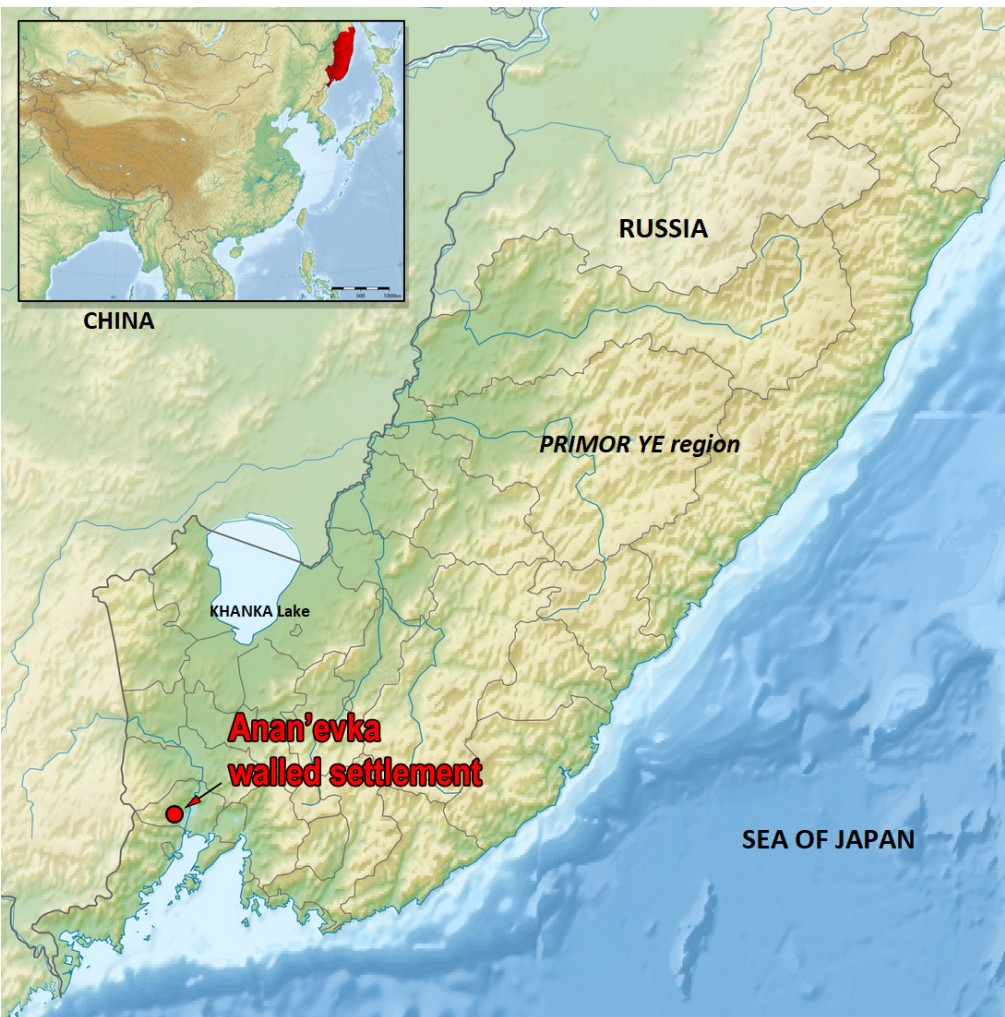

**Figure 1.** The Anan'evka walled settlement location in the territory of the Primor'ye region.

Colors of the figurine's polychrome decoration are red, green, black, and yellow on a white background. This color range has no close similarities among known porcelains and grazed ceramics samples from Jin period sites in the Primor'ye region. Meanwhile, this color range is certainly characteristic for items of "red-and-green porcelain", or "Honglvcai porcelain", first produced in the Cizhou kilns of Northern China in the Jin period. During recent decades, scientific interest in this ceramics category marking such a phenomenon as the appearance of colored overglaze enamel technology in China is constantly increasing [4–9]. However, as the researchers note, this subject is not studied completely [6,7].

The present article considers the results of examination of the polychrome glazed ceramics figurine from the Primor'ye region, southern Russian Far East. We attest to the hypothesis of probable attribution of this art object to the "red-and-green porcelain".

## 2. Materials and Methods

### 2.1. Sample

Our research object consists of several restored parts of the image of a human figure wearing a long robe and standing barefoot on the low pedestal. The human figure and pedestal are hollow inside. The front part of the human body and hands were not preserved. The upper, middle, and lower parts of the figurine are convincingly distinguishable (Figures 2 and 3).

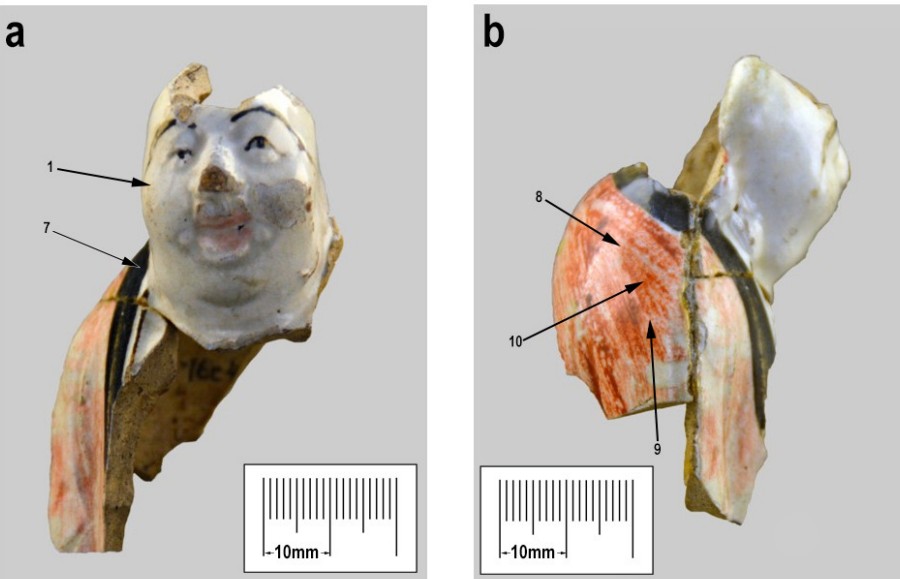

**Figure 2.** Polychrome figurine from Anan'evka walled settlement. Upper part of the figurine: (**a**)—front position, (**b**)—profile position. The arrows correspond to the points of taking the pXRF spectra.

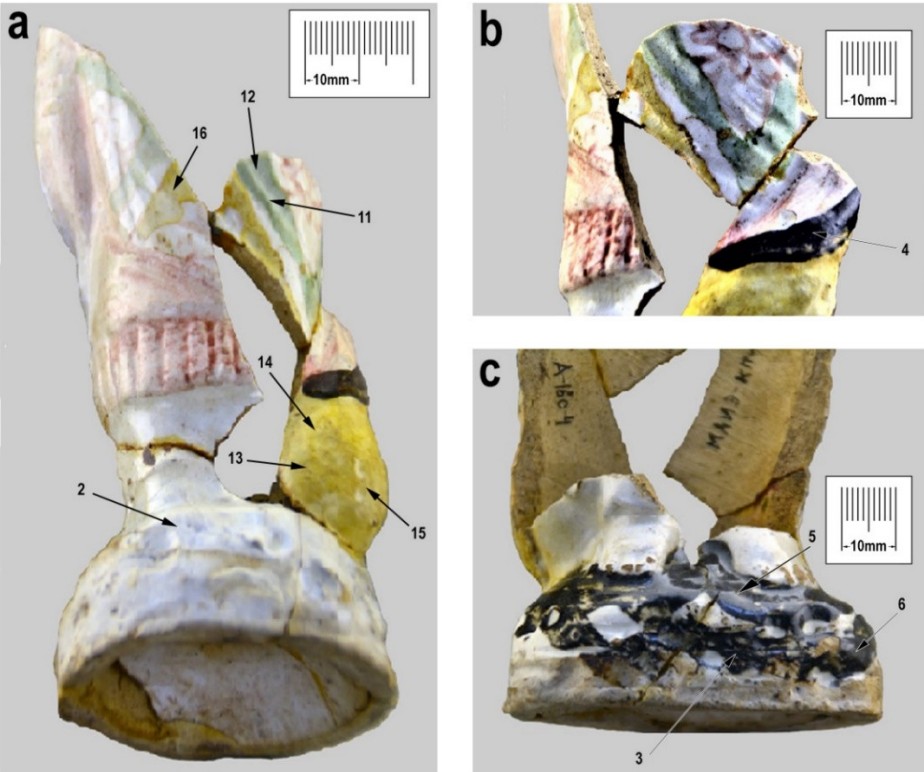

**Figure 3.** Polychrome figurine from Anan'evka walled settlement. Middle (**a**,**b**) and lower (**c**) parts of the figurine. Thin arrows correspond to the points of taking the pXRF spectra.

The total height of the figurine falls approximately within the interval 15–17 cm. The pedestal of 2.3 cm height is formed as a smooth convexity with ellipse-shaped base 4.4 × 6.5 cm. The outer surface of the pedestal has an uneven pitted relief.

Polychrome decoration of the figurine is provided with a white slip and colorful coverings of a glazed type. The human figure's face looks attractive and impressive—the

eyes and brows are carefully drawn in black, the smiling lips tinted slightly with light red. The robe is colored in red and green, with black edging of the collar and flaps. The robe is designed in a pattern of green stripes and red flowers as may be seen in the middle part of the figurine (Figure 3a,b). The front area of the pedestal, under the human figure's feet, is black. On the lower part of the figurine, above the pedestal and a little aside, the feet are a pear-shaped or bag-shaped convexity of 3.0 × 2.5 cm. The surface of the convexity is yellow. On the inner surface of the figurine, flattened areas left by some kind of pressing instrument are visible.

One separate small fragment cannot be fitted to the restored parts of the figurine (Figure 4). The outer surface of the red, black and white fragment is of uneven relief with some elevated and depressed areas (Figure 4a). It seems likely that the colored pattern on this fragment relates to the robe design. On the inner surface of the fragment certain imprints appearing as flattened areas with clearly outlined edges are clearly visible (Figure 4b).

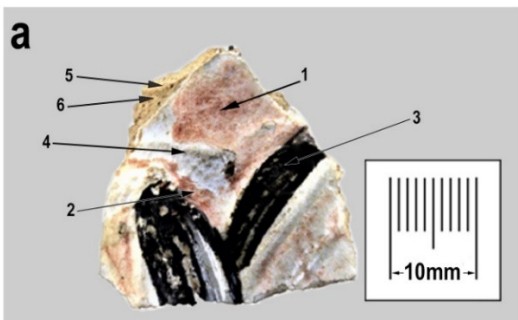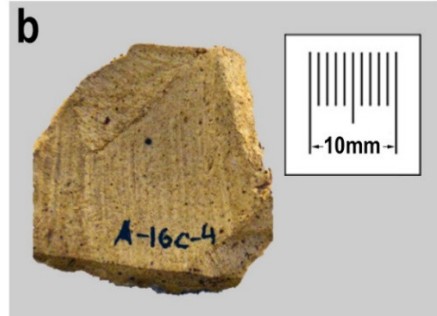

**Figure 4.** Polychrome figurine from Anan'evka walled settlement. Separate fragments. (**a**)—outer surface, the arrows mark sites of SEM-EDS examination. 1—red glaze decoration, point 1; 2—red glaze decoration, point 2; 3—black glaze decoration, point 1; 4—white slip, point 1; 5—ceramics paste, point 1; and 6—ceramics paste, point 2. (**b**)—inner surface with the imprints.

The polychromic figurine from the Anan'evka walled town is stored in the vaults of the Museum of Archaeology and Ethnography of the Institute of History, Archaeology & Ethnology of Peoples of the Far East, Far Eastern Branch of the Russian Academy of Sciences, in Vladivostok city, Russia. The artefact has the status of a unique cultural heritage.

### 2.2. Methods

The research methods applied in our study were purposed to obtain a set of preliminary data on technological characteristics of a polychrome ceramic figurine. The choice of methods used was determined, first, by their non-destructiveness, and second, by technical availability. Analytical methods involving the violation of the object's integrity even in the slightest degree were not considered for the examination because of the cultural strategy of the academic museum where the artefact is stored. The available non-destructive analytical methods were binocular microscopy, optical microscopy, scanning electron microscopy, energy-dispersive spectroscopy, and portable handheld X-ray fluorescence spectroscopy.

Binocular microscopy (BM). A binocular magnifier with 10× magnification was used for diagnosing some macro-traces that could not be seen clearly with the naked eye. This universal instrument provides observations for various objects differing in size and configuration. The BM was applied to the examination of outer and inner surfaces, and the profile section of the restored figurine's parts and separate fragment. The textural features of the ceramics paste, slip layer, and glaze coverings were observed. On the inner surface the imprints supposedly caused by the shaping process were observed.

Optical microscopy (OM). This method was purposed to detail observations of the textural characteristics of the various materials, particularly the glazes [7,10]. We used an

inverted Carl Zeiss Axiovert 40 MAT microscope, assembled with a digital AxioCam ERc 5 s camera (Carl Zeiss Microlmaging GmbH production Germany), with a magnification range of 10–1000×. The equipment is not fitted for work with any objects of complex configuration and highly uneven surface relief. In the case being considered, OM examination was applied to the above-mentioned small separate fragment. Some observations were provided for the profile section of the sample, including layers of ceramics paste, slip, transparent glaze, black glaze, and for outer surface areas of black and red glaze decoration.

Electron scanning microscopy and energy-dispersive spectroscopy (SEM-EDS). The SEM-EDS method was widely adopted for the examination of the substance microstructure and chemical composition in the research strategies in the field of archaeological ceramics and glazes [7,11,12]. The SEM-EDS equipment used in our project was a ZEISS scanning electron microscope EVO-40 (Oberkochen, Germany). The SEM device was fitted with the energy dispersive spectrometer (EDS) of the Oxford Instruments INCA-x sight. The acceleration voltage of SEM examination was kept at 20 kV; the working distance was held at 13.0–16.5 mm; and the magnifications were 346–368×, and 1000×.

The sampling box of used SEM-EDS equipment is designed for the examination of relatively small and simple shaped objects. Only individual fragments of the polychromic figurine were appropriate to be analyzed with this equipment. The sample was prepared by carefully removing dust pollution from the surface with a tissue wet with ethyl alcohol. SEM-EDS examination was applied to surface areas. No special films or coatings (carbon, chromium, gold) were applied to the sample before examination. SEM images and taking of EDS spectra were recorded for 2 sites (points) of ceramics paste, 2 sites (points) of red glaze decoration, 1 site (point) of white slip and 1 site (point) of black glaze decoration. The amounts of spectra taken for each site (point), were from 9 to 21 ones. The locations of SEM-EDS sites (points) are marked in Figure 4a. Taking into account that the EDS method determines the content of chemical elements (in % of the weight composition) at a semi-quantitative level, values from 1.0% and higher were most considered for the correct interpretation of elemental compositions data. The results of elemental compositions measurements are presented in Table 1 summarizing the mean values of chemical elements and box plots (also known as "box-and-whiskers" diagrams) allowing the evaluation of the degree of homogeneity or heterogeneity of the elemental compositions.

**Table 1.** EDS data on the major chemical elemental composition of the paste, slip, black glaze, and red glaze tested on a separate fragment of the figurine (mean values).

| Sample | Point No | Spectra Amount | C | O | Al | Si | K | Ca | Ti | Fe | Pb |
|---|---|---|---|---|---|---|---|---|---|---|---|
| Paste | 1 | 9 | 20.22 | 43.13 | 11.40 | 11.33 | 1.04 | | 4.58 | 7.21 | |
| | 2 | 20 | 15.39 | 45.17 | 12.23 | 14.37 | 1.50 | 0.63 | 2.28 | 1.41 | |
| Slip | 1 | 9 | 33.01 | 38.82 | 9.76 | 12.13 | 1.12 | 2.01 | | | |
| Black glaze | 1 | 13 | 12.16 | 36.24 | 2.95 | 19.69 | 1.45 | 1.14 | | 1.70 | 23.21 |
| Red glaze | 1 | 13 | 33.30 | 29.71 | 2.59 | 14.25 | 1.43 | 1.16 | | 2.61 | 14.05 |
| | 2 | 21 | 20.17 | 36.74 | 3.31 | 19.01 | 1.53 | 1.40 | | 2.03 | 14.78 |

Portable X-ray fluorescence spectroscopy (pXRF) allows analyzing surface areas up to 10 mm in diameter on objects of any shape and size and providing approximate average quantitative data on the content of chemical elements (in % of the weight composition). In modern archaeometry, there is the experience of pXRF being applied in the research of glazed ceramics [13,14]. In our project, the portable X-ray fluorescence analyzer model Olympus Delta Professional DP 4000 is used to study the glaze coatings on the restored parts of the statuette, which is inaccessible for analysis with the EDS unit (Figure 2a–c). The analyzer was calibrated with the standard Olympus Analytical Instrument 316 calibration check reference coin. The operating specifications of this pXRF analyzer determined the limitations in the diagnostic capabilities of some light chemical elements whose atomic

weight is less than 44, in particular, concerning Ca and alkali elements K, Na, Mg which are usual components in the structure of ceramic masses and glazes. The pXRF-analyzer allowed for obtaining information about the content of the composition of glazes' basic elements Si and Al, as well as elements-metals that are of interest as probable color-forming components (Fe, Cu, Mn, Cr, etc.) and fluxing additives (Pb).

Measurements were taken in an air atmosphere through a PRO 6 Prolen U8990460 (6 μm) window, in dual beam mode for 40 s (10 s, 30 s). The spectra measurements were taken at 16 points corresponding to the object's surface areas with transparent, black, red, and green glaze coverings. The points are marked in Figure 3. The results are presented in Table 2. In order to interpret the obtained data correctly, chemical elements with values ≥1.0% (wt) were taken into account.

**Table 2.** pXRF data on the chemical elemental composition of glazes.

| Sample | Point No. | Chemical Element, wt.% | | | | | | | | | | | | | | | | |
| | | Si | Fe | Al | P | Ti | Mn | Zr | Zn | S | Cu | Nb | Pb | Sb | Sn | Cd | Bi | Hf |
| Transparent glaze | 1 | 75.42 | 3.19 | 15.20 | 0.32 | | 0.17 | 0.22 | | 5.19 | 0.07 | 0.03 | 0.17 | | | | 0.03 | |
| | 2 | 75.43 | 2.59 | 16.41 | 0.49 | 0.43 | 0.08 | 0.19 | | 4.11 | 0.04 | 0.03 | 0.17 | | | | 0.02 | |
| Black glaze | 3 | 67.75 | 14.02 | 14.57 | 0.44 | 0.29 | 0.66 | 0.31 | 0.07 | 1.41 | 0.11 | 0.04 | 0.28 | | | | 0.05 | |
| | 4 | 62.96 | 6.28 | 14.72 | 0.53 | 0.32 | 0.33 | 0.39 | | | | 0.06 | 14.41 | | | | | |
| | 5 | 72.35 | 7.74 | 14.33 | 0.83 | | 0.19 | 0.38 | | | | 0.07 | 3.82 | | | | | 0.29 |
| | 6 | 72.13 | 8.62 | 16.92 | 0.53 | 0.26 | 0.34 | 0.23 | | | | 0.04 | 0.86 | | | | | 0.08 |
| | 7 | 72.58 | 7.49 | 16.60 | 0.63 | 0.43 | 0.24 | 0.29 | | | | 0.05 | 1.56 | | | | | 0.13 |
| Red glaze | 8 | 74.21 | 5.32 | 16.51 | 0.97 | 0.42 | 0.15 | 0.29 | | | | 0.05 | 1.93 | | | | | 0.15 |
| | 9 | 73.52 | 6.70 | 15.92 | 0.88 | | 0.18 | 0.41 | | | | 0.06 | 2.12 | | | | | 0.22 |
| | 10 | 75.98 | 4.98 | 15.34 | 0.92 | 0.35 | 0.17 | 0.29 | | | | 0.05 | 1.77 | | | | | 0.15 |
| Green glaze | 11 | 37.49 | 3.90 | 9.64 | 0.35 | 0.25 | | 0.43 | | | 0.41 | 0.06 | 47.01 | 0.46 | | | | |
| | 12 | 46.62 | 2.90 | 11.41 | 0.49 | 0.32 | 0.09 | 0.56 | | | 1.39 | 0.08 | 36.15 | | | | | |
| Yellow glaze | 13 | 30.77 | 2.73 | 3.19 | 0.28 | | | 0.10 | | | | | 61.54 | 0.36 | 0.18 | 0.12 | | 0.73 |
| | 14 | 30.41 | 2.51 | 3.23 | 0.27 | | | 0.08 | | | | | 62.37 | 0.43 | 0.17 | | | 0.53 |
| | 15 | 42.85 | 2.49 | 4.62 | 0.23 | | | 0.11 | | | | | 48.66 | 0.38 | 0.19 | | | 0.47 |
| | 16 | 68.14 | 3.02 | 13.41 | 0.82 | | 0.12 | 0.62 | | | 0.89 | 0.08 | 12.90 | | | | | |

Water absorption (WA) testing. It is well known that ceramics water absorption, or relative porosity, is a physical property caused by the firing temperature regime and the degree of ceramics paste vitrification. Usually the water absorption value (in %) is determined by the accounting ratio of the difference between the dry ceramic sample weight and the water saturated sample weight to the dry sample weight [15]. This testing method was applied to the separate fragment to obtain additional data on the ceramics body quality.

## 3. Research Results

### 3.1. Ceramics Paste

The ceramic body of the figurine is 0.3–0.6 cm thick and an even brownish-yellow color. The above-noted imprints on the inner surface look similar to traces of a spatula-like instrument moving over the plastic clay mass. An instrument of about 2.0 cm width had a sharply slanting edge (Figure 4b). Seemingly, these imprints were left during the process of shaping the figurine.

The BM and OM show that the ceramics paste has a dense but not homogenous texture (Figure 5a).

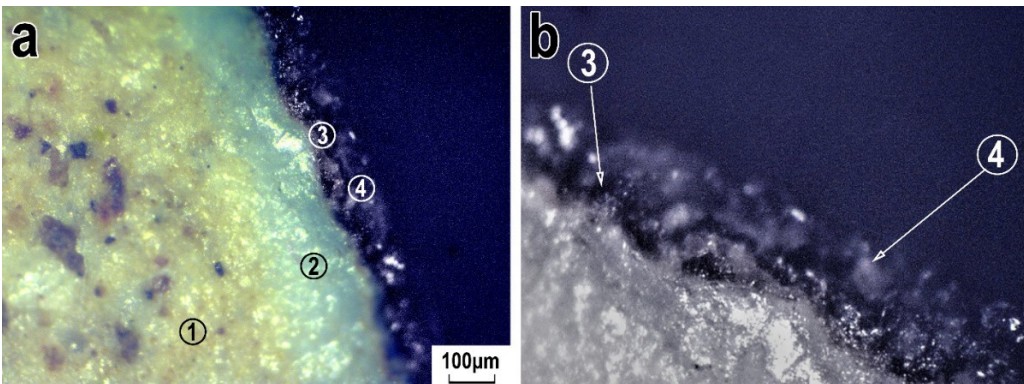

**Figure 5.** The OM image of the sample's profile section showing layers sequence. (**a**): 1—the paste, 2—white slip, 3—black glaze, 4—transparent glaze. (**b**): 3—black glaze, 4—transparent glaze. The magnification 50×.

The plastic clay fraction and non-plastic mineral fraction are distinguished. The quartz, feldspar, and iron oxides particles are recognized in non-plastic fraction. Mineral grains distributed evenly are mostly of ≤0.3–0.2 mm in size and predominantly of angular contours. The grains of 0.5–1.0 mm are single ones. In general, judging from their features, the mineral inclusions may be interpreted as natural clay temper rather than artificial additives [16].

According to EDS data, the paste chemical composition shows almost equal concentrations and a range of values of Al и Si (Table 1, Figure 6a). The Al:Si ratio accounted based on the mean element weights is 1.00 for the point 1, and 0.85 for the point 2. As is known, the balance between alumina and silica, or $Al_2O_3$ and $SiO_2$, is quite important in influencing certain properties of clay materials. In particular, the higher the alumina content, the higher the clay refractoriness and, consequently, firing temperature [17,18]. The concentrations of Fe and Ti differ significantly at points 1 and 2 (Table 1). The paste composition at point 1 is characterized by great dispersion of the Fe concentration (minimum ~1%, maximum ~40%.) which indicates essential heterogeneity of the iron distribution in the clay material (Figure 6a). The content of Ca is too low for any probable impacting of the paste properties. It may be said that raw material used for the ceramic figurine was high-alumina non-carbonaceous clay with good refractory properties. Seemingly high iron and titanium as coloring agents caused the brownish-yellow fired paste color.

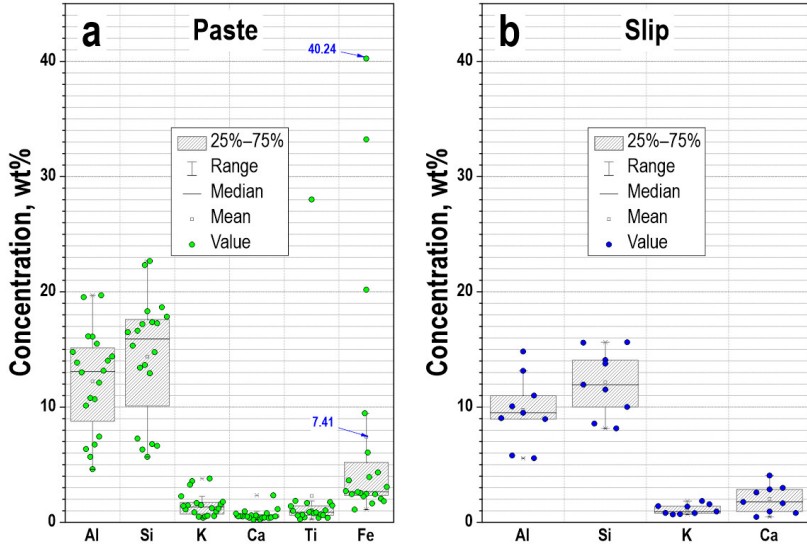

**Figure 6.** Box-and-whiskers plots for chemical compositions of paste (**a**) and slip (**b**).

SEM examination of the separate figurine fragment revealed evidence of developed, or extent, vitrification of the paste substance—glass-like melted areas and clusters of smallest "closed" roundish pores (Figure 7).

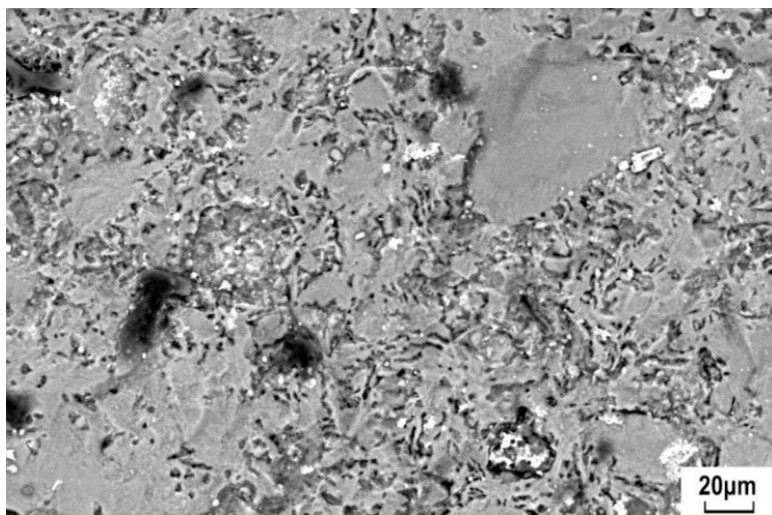

**Figure 7.** SEM image of ceramic paste texture with evidence of developed vitrification. The magnification 368×.

The vitrification in non-carbonaceous low refractory clays begins at 800–850 °C under oxidizing atmospheric conditions and achieves a developed stage at 900–950 °C. In high refractory clays the process of vitrification does not begin until the clay is fired at 1000 °C [12]. The WA index determined for this fragment is 0.7%—the value may be considered as very low, taking into account that the WA index for a modern stoneware body is ≤2.0% [19].

*3.2. Slip*

A white slip layer 0.2–1.0 mm thick totally covers the outer surface of the figurine and its pedestal's inner surface. Based on the BM and OM examinations, the slip does not contain any non-plastic inclusions similar to the paste inclusions. The slip and paste layers demonstrate obvious textural and color differences (Figure 5a). According to the EDS data the Al concentration in the slip substance is relatively close to the Si concentration (Table 1). The Al: Si ratio is 0.8. Dispersion of Al and Si concentrations in the slip composition shows a more homogenous pattern in comparison with the paste composition (Figure 6b). In contrast to the paste composition, the Fe and Ti concentrations are quite low, at a few tenths of a percent, while the Ca is present at 2.01% (Table 1).

At examined point 1, a single EDS spectrum No. 9 detected a high concentration of zirconium as 13.86%. The SEM image of the point 1 shows that this spectrum corresponds exactly to a brightly white angled grain of a metallic type (Figure 8). As is known, the zirconium occurs in some clays as natural mineral inclusions or impurities [20]. The presented data may be interpreted as just such a case.

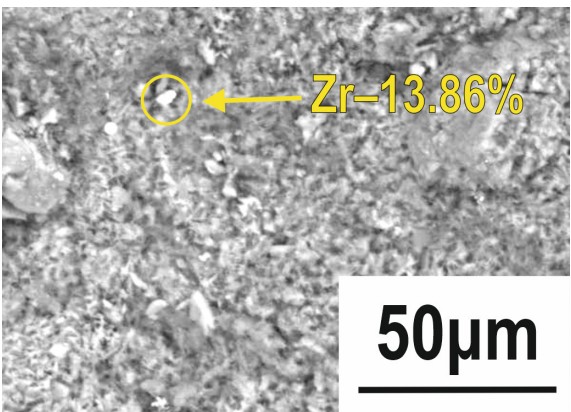

**Figure 8.** SEM image of the spectral surface with the indicated location of the zirconium inclusion.

### *3.3. Glazes*

### 3.3.1. Transparent Glaze

This shining glaze with glass-like texture appears directly above the white slip on the figurine's face, head, and neck areas, and on some areas of the pedestal (Figures 2 and 3a). Under the BM a crackling net-like pattern with meshes from $1.5 \times 1.5$ mm to $2.0 \times 3.0$ mm in size is clearly visible. The OM of the separate fragment revealed a profile section where a transparent glaze layer of up to 0.15–0.2 mm thick is lying above a very thin layer of black glaze (Figure 5a,b). The pXRF detected a high content of the Si and low content of the Al. The ratio Al:Si is 0.20–0.21. The presence of Fe and S (sulfur) may be noted (Table 2). The sulfur, supposedly, may be an impurity that penetrated into the glaze from waste gas during the firing process [21].

### 3.3.2. Black Glaze

Visually the black areas of the figurine look glossy and smooth like glass. The OM observations on the separate fragment revealed an intermediate position of black glaze layer of ≤0.1 mm thick—above the white slip layer and under the transparent glaze layer (Figure 5a,b).

The texture of the black areas on the separate fragment was examined with the OM and SEM. In the OM image of the horizontal plan multiple thin scratch-like crackles are overlapped above with single deeper wide crackles and clusters of small bubbles associated seemingly with the transparent glaze layer (Figure 9a). The SEM examination detected large deep crackles of 0.01 mm wide corresponding to the overlying transparent glaze and micro-crackles forming a crystallization pattern. The micro-crackles cut with deep wide crackle lie under the transparent glaze corresponding to the black glaze layer (Figure 10a).

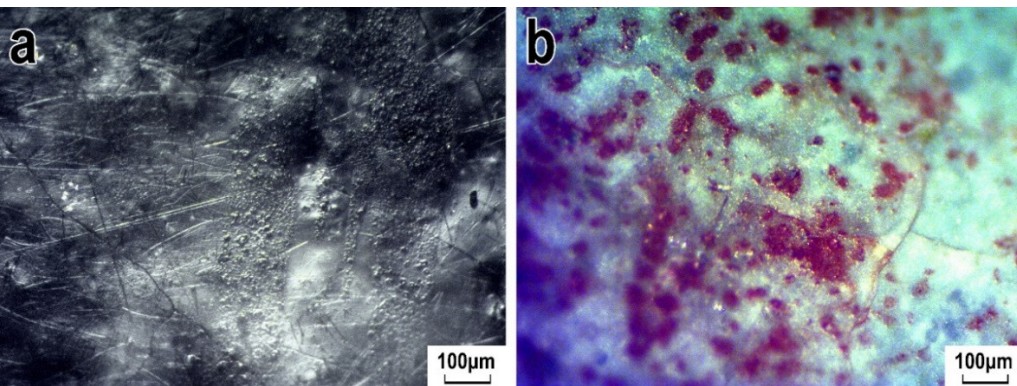

**Figure 9.** OM images of surface texture of the black area (**a**) and red area (**b**) on the separate fragment of the figurine. The magnification is 50×.

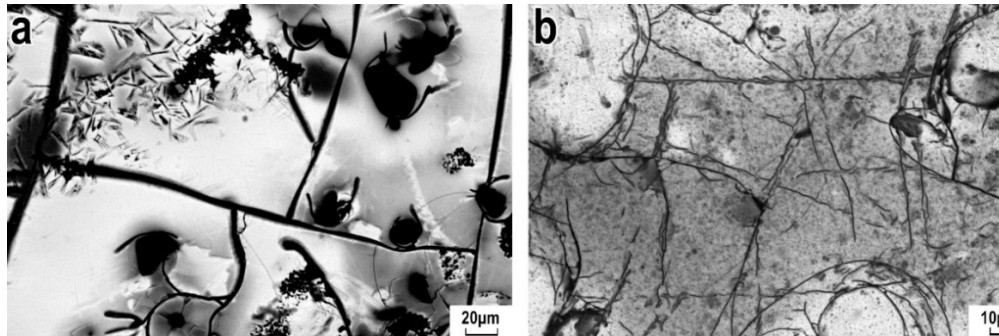

**Figure 10.** SEM images of surface texture of the black area (**a**) and red area (**b**) on the separate fragment of the figurine. The magnification is 346×, 368×.

The elemental composition of the black glaze was tested by the pXRF at four points in some surface areas of the restored figurine's parts (Figures 2a and 3b) and by the EDS at one point on the separate fragment (Figure 4a). Obviously, the obtained data reflect not only the composition of the black glaze but also the composition of the above-layered transparent glaze (Tables 1 and 2). The strongly dominating element in the pXRF and EDS spectra is Si. The Al: Si ratio is 0.19–0.23 according to the pXRF, and 0.13 according to EDS. High amounts of Pb were detected by both methods. This element may undoubtedly be associated with the black glaze composition but not with the composition of the transparent glaze lying above.

It is possible to make some interesting observations on the lead concentration at various points by taking the pXRF and EDS spectra. Based on the pXRF data, the highest concentration, 14.41%, was detected at point 4, located very close to the yellow glaze decoration area (Figure 3b). Minimal and insignificant concentrations, 0.28% and 0.86%, are detected at points 3 and 6, located in the low area of the figurine's pedestal, apart from contact with the colored glaze decoration areas (Figure 3c).

The Fe concentration in the pXRF spectra is 6.28–14.02% and 0.40–3.53% in the EDS spectra. As is known, the presence of ferric oxides in glaze composition activates the crystallization processes during post-firing cooling [11]. Ca and K in the EDS spectra are presented in insignificant and almost equal concentrations and dispersions (Table 1, Figure 10a). The plot diagram of black glaze element concentrations shows significant heterogeneity of the Si and especially Pb in contrast to the Al, K, Ca, and Fe (Figure 11a).

### 3.3.3. Red Glaze

Red was the dominant color of the robe decoration. Red colored covering is of uneven saturation on different parts of the figurine. At the back and shoulder area of the upper part the color is relatively bright and intensive, but in other areas it looks blurred and dull. The red covering was applied with a brush—traces of brushing are seen at the surface of the figurine's back (Figure 2b).

The OM of the red area on the separate fragment revealed a texture that appeared as drops or "islets" of various size and shape located above the transparent glaze layer (Figure 9b). Under the BM, some cases of transparent glaze crackles filled with red pigment may be observed. The SEM image of red glaze reveals micro-crackles of straight and slightly curved contours (Figure 10b).

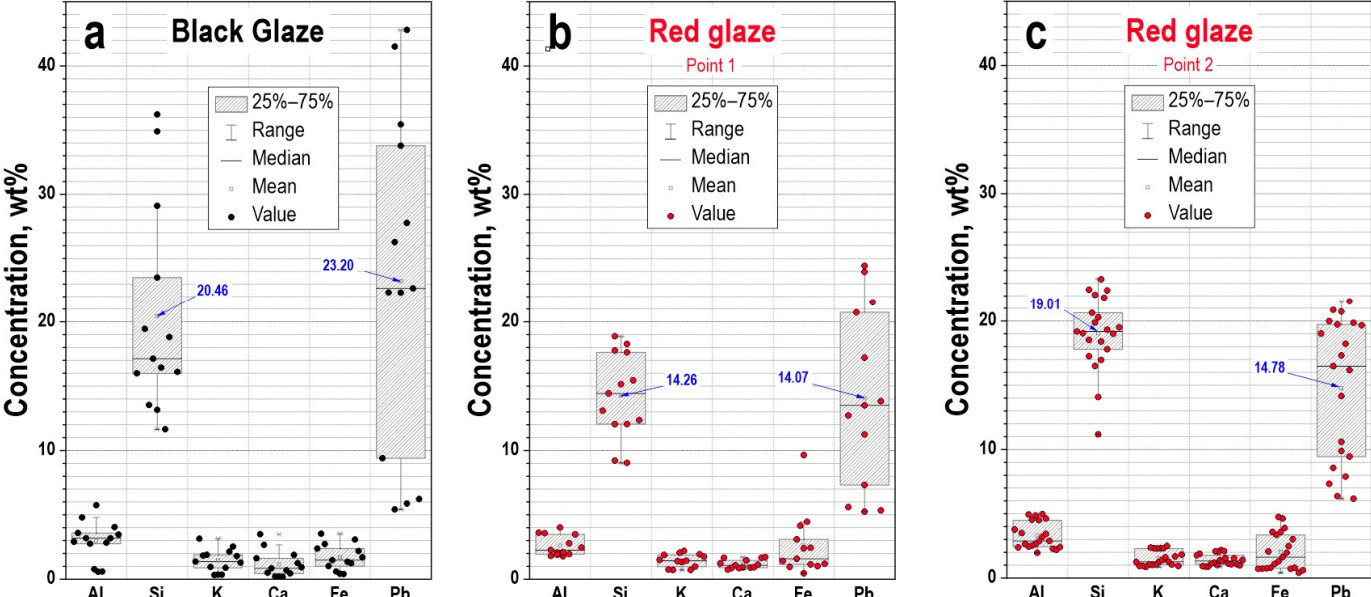

**Figure 11.** Box-and-whiskers plots for chemical compositions of black glaze (**a**) and red glaze at point 1 (**b**) and point 2 (**c**).

The EDS and pXRF spectra indicate a significant content of Si in the chemical composition (Tables 1 and 2). The Al:Si ratio is 0.18 according to EDS data for points 1 and 2, and 0.20–0.22 according the pXRF data. Both methods indicate relatively high concentration of the Fe. EDS spectra in points 1 and 2 detect a relatively high concentration of the Pb while in the pXRF spectra the lead concentration is insignificant. According to the EDS, concentrations of Ca and K are close to their concentrations in the black glaze.

The plots of red glaze at point 1 and point 2 shows very close dispersion patterns for all elements (Figure 11b,c). At the same time, some differences between element dispersion patterns of black glaze and red glaze may be noted. In black glaze all elements show more pronounced dispersion in comparison with red glaze. The most notable difference is between the Pb dispersion patterns in black and red glazes. Supposedly, these differences indicate generally greater homogeneity of the elemental composition of the red glaze compared to the black. This is especially evident for the Pb which shows much greater heterogeneity in black glaze.

### 3.3.4. Green Glaze

This semi-opaque, weakly shining glaze was used for the coloring strip-like pattern on the robe (Figure 3a,b). The pXRF examination revealed a high content of Si and low content of Al. The Al:Si ratio is 0.24–0.25. A very high content of the Pb, up to 47.01%, was detected. Among the elements with weight values ≥1.0% are recognized Fe and Cu (Table 2).

### 3.3.5. Yellow Glaze

A bright yellow opaque glaze covers the bag-shaped convexity above the pedestal near the anthropomorphic person's feet (Figure 3a). Below, Figure 12 shows yellow on the glazed surface.

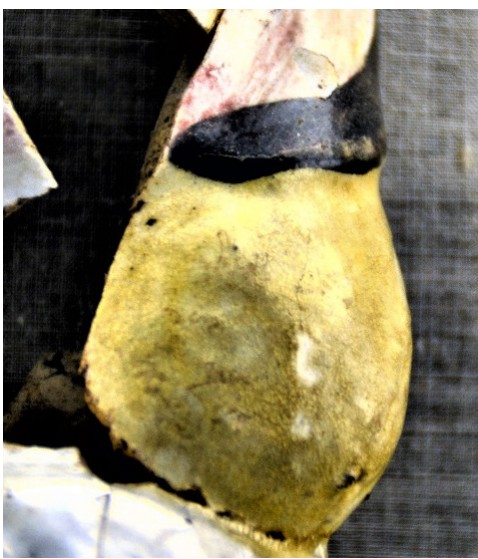

**Figure 12.** Polychrome figurine from Anan'evka walled settlement. Yellow glaze is on the surface of the bag-shaped convexity.

Under BM a fine-meshed net of very thin crackles is visible. The pXRF examination determined a high content of Si and low content of Al, with an Al:Si ratio of 0.10–0.19. The very high Pb content, up to 62.37%, is characteristic for the elemental composition, as in the case of the green glaze. The Fe concentration is 2.52–3.02%, which is close to the green glaze. The presence of Sb (antimony) in the concentration is less than 1.0% and is detected in three spectra out of four (Table 2).

## 4. Discussion

The results of technological examination suggest some consideration of the process of making the polychrome figurine. Such features as relatively thin paste texture, thin walls of the hollow figurine corpus, and traces of a spatula-like instrument on the inner surface indicate with high probability the use of a mold for shaping the figurine. It seems likely that the mold was composed of two sections, or halves, a front and a back. A plastic clay mass was pressed into the mold's sections with a spatula-like tool. After partial drying the formed parts of the figurine had to be removed from the mold's halves and joined together. It may be noted that the technology of shaping figure-like objects by pressing thin slabs of clay into hollow molds was widely practiced in China from the end of 1 mil. BC. In particular, sectioned molds were used for shaping complicated objects [22,23] (pp. 424–430).

The slip layer had to be applied to the figurine when its outer surface was still plastic and sufficiently wet. The white slip served as the make-up of the brownish-yellow body and background for the decoration of polychromic glazes. Chemical compositions of slip and clay paste are characterized by almost equal values of the Al: Si ratio. This indicates, first, that the same degree of paste and slip plasticity that is important for strong bonding and even drying of these two substances. The high Al content in the paste and slip compositions indicates their appropriateness for high-temperature firing.

Based on the EDS data, the paste and slip compositions differ in the presence of Fe and Ti. In contrast to the slip, the paste composition contains these elements. The researchers emphasize that the usage of titanium-containing clays was practiced widely in Northern China for stoneware production. However, raw clay materials used for true porcelain production did not contained titanium [23] (p. 473). Supposedly, some kind of clay similar in certain properties to porcelain raw material was used for the slip.

The elemental compositions of transparent and colorful glazes have a high silica content with Al: Si ratio varying from 0.13 to 0.23. The basic compound of the glazes

seems to be raw material rich in silica such as quartz, feldspar, or quartz-feldspar sand [8]. Transparent colorless glaze is a non-lead one, while all colorful glaze coverings contain lead in various concentrations.

Based on the analytical data a certain sequence of the glazing process and firing treatment may be supposed. The results of OM and BM examination allows us to think that the black glaze outlining the facial details, robe's edging, and front pedestal area was applied above the white slip and then covered by a transparent glaze. The first firing seems to have been executed after the transparent glaze was applied. The firing was conducted at a high temperature in an oxidizing regime, judging by the evidence of developed vitrification and the brownish-yellow color of the ceramics paste, and by the evidence of crystallization in the black glaze. In this connection, such a feature as the presence of lead in the black glaze composition is detected by the EDS spectra and some pXRF spectra is noteworthy. The lead seems to be an inconsistent element in the glaze intended for the high-temperature firing. An explanation for this seeming curiosity will be suggested below. Lead-containing red, green, and yellow glazes were applied under the transparent glaze, and then the figurine was fired at a low temperature.

According to the EDS and pXRF data on the black and red glazes, the iron is supposed as the main colorant. Iron-containing glazes of a broad color range depending on the mineral phases diversity and atmospheric firing conditions were quite popular in ceramics production in ancient China [4] (pp. 159–166); [24]. Copper was recognized together with iron in the green glaze. The copper oxide in lead glazes fired under oxidizing conditions results in the appearance of green colors [13,15] (p. 338).

As revealed, the elemental composition of yellow glaze contains iron in a concentration of 2.49–3.02%, and antimony (Sb) in a concentration of less than 1.0%. The antimony is known as a component for producing yellow glazes [13,23] (pp. 470–471). In our case this element may be preliminarily supposed as some kind of colorant together with the Fe.

The noted technological traits seem to have certain similarities with the chaîne opératoire of "red-and-green porcelain", or Honglvcai, as a special category of ancient Chinese ceramics. Its provenance area is associated mostly with Cizhou kilns system located in Northern China. including such well-known workshops as Guangtai and Linshui in the Hebei province, Pacun and Hebiji in Henan province, Jiexiu and Bayi in Shanxi province, and others (Figure 13). The Song-Jin period was a time when cluster Cizhou kilns flourished, producing many popular sorts of ceramics. Different kilns shared common technological and artistic principles, receipts, and trends of ceramics production activity [4] (pp. 229–232); [7]; [23] (pp. 170–172); [25]. It is accepted generally that the technology of overglaze colored enamels and the polychrome Honglvcai originated in the Cizhou kilns in the mid-Jin period, around 1200 CE [4] (p. 230); [6]; [23] (p. 170; 616); [25]. The researchers consider Honglvcai as the prototype or technological "ancestor" of polychrome Doucai and Wucai porcelains with overglazed enamels—brilliant innovations of the Ming period (1368–1644 CE) [6,7].

The "red-and-green" ceramics were produced mostly of stoneware clays but not of true porcelain raw material. The researchers note the yellowish-grey color of the Honglvcai body, characterized by relatively high alumina content and some amount of iron. The composition of the white slip applied to the body also contains high alumina (Tables 1 and 2 in [7]). Stoneware clays rich in alumina and intended for high temperature firing, up to 1200 °C and more, were used widely in Cizhou kilns pottery production. The fired body was an off-white or creamy color that improved the application of a white slip. The usage of white high-alumina slip was one of the distinctive technological trends of the Cizhou kilns [23] (pp. 176–178).

Obviously, the ceramics' paste and slip characteristics obtained for the polychrome figurine examined are consistent with these data.

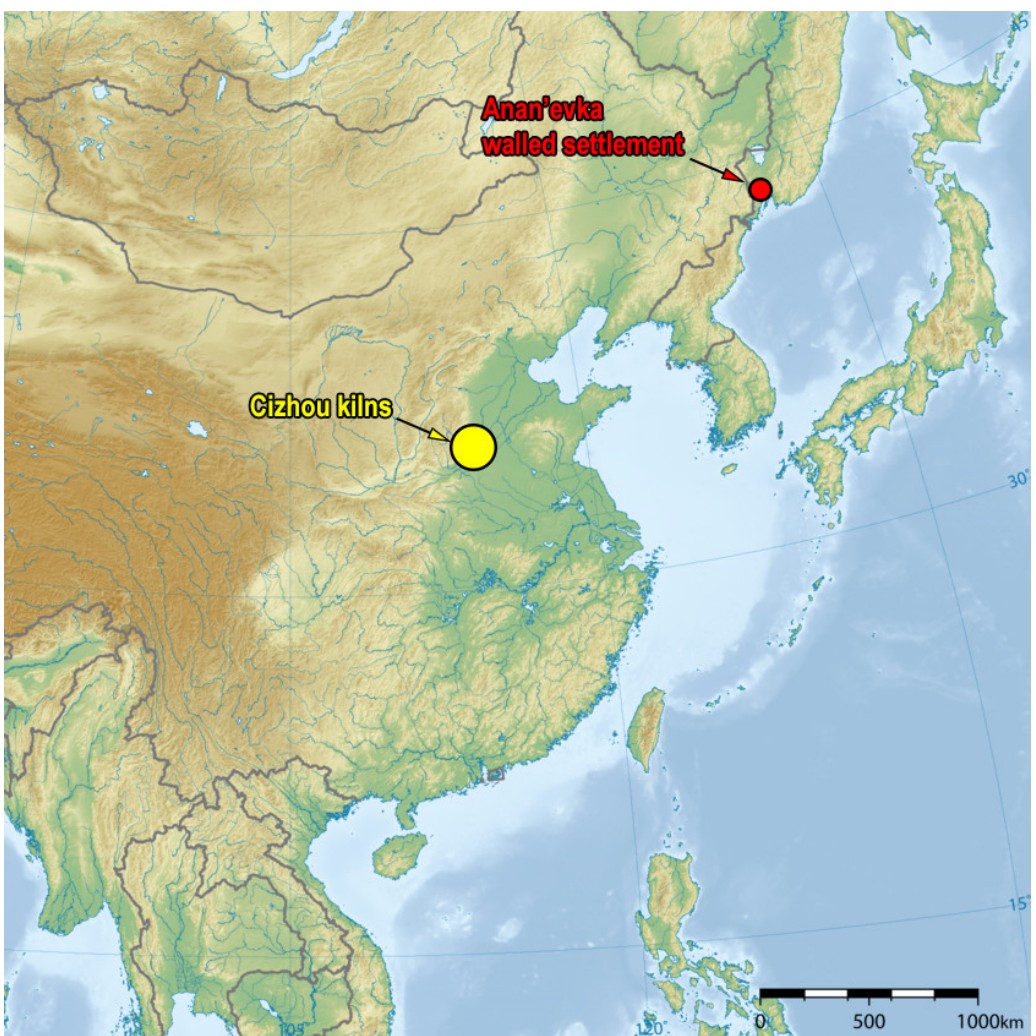

**Figure 13.** Location of the Cizhou kiln and Anan'evka walled settlement at the map of the Asia-Pacific Region.

Underglaze decoration was produced by drawing black contour lines and fine details on a white slip [9]. The case of a special study of the black decoration of the Honglvcai sample excavated at the Linshui kiln site was presented recently by X. Jiang and co-authors [7]. The examined sample's decoration colors are red, green, yellow, white, and black. The black decoration chemical composition is characterized by a relatively high concentration of iron while the concentration of manganese is very low. The PbO varies in the range of 5.05% to 36.78% at four measurement points. The researchers emphasize significant lead content in the black decoration composition, 22% on average which corresponds usually to low temperature glaze. However, it is argued that black decoration is a high temperature one. Due to its extremely high flowability and volatility the Pb had penetrated into the black decoration layer from a nearby area of lead-containing yellow overglaze enamel through the underlying transparent glaze during secondary low-temperature firing and after-firing cooling [7].

The referenced data seem to be quite important for correct understanding of some results of our study, in particular the high lead concentration in the composition of the polychrome figurine's underglaze black decoration. As noted above, in Section 3.3.2, high lead content is detected by pXRF and EDS at those points of the black glaze decoration that are located close to the yellow glaze and red glaze decoration areas. In contrast, pXRF spectra taken in the black decoration areas did not contact the colored glazes and showed minor values of lead (Table 2). As was determined, the lead in the black glaze composition

showed a high degree of heterogeneity (Figure 10a). Taking into account these observations, it seems probable that the presence of lead in the black glaze composition can be explain by its infiltration from overglaze colored coverings during or just after secondary low temperature firing.

Red and green as dominant colors of Honglvcai decoration were produced by lead-containing low temperature enamels. They were applied over cooled transparent glaze, and the ceramic object was then fired at a temperature around 800 °C for a short time [8,23] (p. 615). The red enamel coloring agent was determined to be ferric oxide, in particular, hematite crystallite [4] (pp. 230–231); [6]. In the green enamel, copper is detected as the colorant [4] (p. 231); [8]. The data on the probable colorants of red and green glazes on our polychrome figurine do not contradict previous determinations.

Yellow played the role of local accent in the color scheme of the Honglvcai. Yellow enamel may have been used sometimes to emphasize certain details of a whole decorative image [9]. Considering the Cizhou glazes and enamels, N. Wood in 1999 noted the absence of accurate analytical data on the yellow enamel chemical composition. He submitted the idea that yellow colorants were ferric oxide in a solution of around 3.0–4.0%. Moreover, he did not exclude the presence of antimony (Sb) in yellow enamels [4] (p. 231). According to the above-mentioned research case of the Honglvcai, ferric oxide was detected in yellow enamel composition (Table 1 in [7]). As for the antimony, this element has been known in the composition of Chinese glazes since the Tang period. The antimony and ferric oxide were detected as components of yellow glazes on some Sancai potteries unearthed at the Huangpu and Huangye kilns of that time. The concentration of antimony is significantly less than the ferric oxide concentration [26]. The results of the pXRF examination of the yellow glaze, or enamel, on the considered polychrome figurine allow us to say that the subject of identifying antimony as the probable component of yellow enamels on "red-and-green" ceramics seems to be a prospect for further investigation and discussion.

The assortment of Cizhou kilns "red-and-green" ceramics included various wares, pillows, and small-scale sculptures [7] (p. 25). The sculptures shaped by molding technology have presented various images of Chinese mythology and religious beliefs, and zoomorphic and anthropomorphic symbols. In particular, the personages of the Buddhism circle were performed in the sculptural miniatures of "red-and-green" decoration style [27]. The researchers emphasize the high artistic level of the "red-and-green" plastic miniature influenced by painting and sculpture arts of the Northern Song (960–1115 CE) and Jin periods. In particular, careful working out of the facial features with fine black lines on white engobe may be considered as an imitation of ink painting on paper or silk. The color scheme of "red-and-green porcelain" sculptural miniatures corresponds to the colors of painting, wood, and clay sculpture creations [9].

The polychrome figurine unearthed in the Primor'ye region may be considered as the example of "red-and-green" sculptural miniature art. Certain iconographic features of the figurine are important for a personal interpretation of the anthropomorphic image. These are a smiling face with plump cheeks, fat neck, standing position, undressed feet, a pedestal looking like some kind of rockery, yellow bag-like convexity near the feet. In general, these features have certain similarities in the iconography of a certain image that became popular in Chinese religious culture, folklore, and arts beginning in the Song dynastic period. This is Budai—the travelling Zen monk, a real person who lived in the tenth century. There are many iconographic variants of the Budai image, and one of them is known by its specific attribute—the canvas bag. The Budai image as the symbiosis of official Buddhism ideas and folk beliefs was symbolized as happiness and fortune [28,29] (pp. 391–392). So, our interpretation is that the considered polychrome figurine presents an image of the monk Budai.

## 5. Conclusions

As the main research result, it may be concluded with high probability that the polychromic ceramic figurine found at the Jin period site Anan'evka walled town in the

Primor'ey region of the southern Russian Far East belongs to the category of "red-and-green porcelain", or Honglvcai. The main technological features and properties of this example, determined by analytical methods, certainly correspond to the distinctive characteristics of "red-and-green porcelain" objects produced in the workshops of the Cizhou kilns system in Northern China as early as the mid-Jin period. It can be argued that the polychrome ceramic figurine portraying the monk Budai may be considered today as the most northeastern case of "red-and-green porcelain" discovered in an archaeological context.

**Author Contributions:** Conceptualization, I.S.Z.; methodology, I.S.Z. and I.Y.B.; investigation, I.S.Z. and I.Y.B.; writing—original draft preparation, I.S.Z.; writing—review and editing, I.Y.B.; visualization, I.Y.B.; project administration, I.S.Z. All authors have read and agreed to the published version of the manuscript.

**Funding:** The research was financially supported by the Russian Science Foundation (project No. 20-18-00081).

**Institutional Review Board Statement:** Not applicable.

**Informed Consent Statement:** Not applicable.

**Data Availability Statement:** Not applicable.

**Acknowledgments:** The authors are grateful to Li Weidong (Shanghai Institute of Ceramics, China Academy of Sciences) for her preliminary consulting on the "red-and-green porcelain" as a specific category of ancient Chinese ceramics. The SEM-EDS examination of the object was executed in the Electron Microscopy Centre of the National Scientific Marine Biology Centre, Far Eastern Branch of Russian Academy of Sciences. The authors thank the Electron Microscopy Centre Denis V. Fomin for the technical assistance and support. The authors are grateful to Richard L. Bland (University of Oregon, Eugene, OR, USA) for his kind assistance with the English corrections and for helpful edits and comments.

**Conflicts of Interest:** The authors declare no conflict of interests.

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
