# Peer review of "A “Red-and-Green Porcelain” Figurine from a Jin Period Archaeological Site in the Primor’ye Region, Southern Russian Far East"

_ceramics, doi:10.3390/ceramics5040049_

Round 1
Reviewer 1 Report
The manuscript, entitled “Red‐and‐green porcelain” figurine from Jin period archaeological site in Primor’ye region, southern Russian Far East” examines only one object and is not compared scientifically with the other Honglucai objects. The interpretation of the data is completely incorrect. For example, for the yellow color, EDS analyses show the presence of Naples yellow, which was not discussed in the text. Black glaze contains iron and manganese, which is different from the red glaze's composition. The content of Al and Si is important for the body analysis. Slip layer contains a significant amount of zirconium which can not be assumed as an impurity. An impurity refer to a concentration less than 1%. For the glaze, alkali content (Na, K, and Mg) may be compared to the content of Pb, not Al and Si. The presentation of the data in Table 1 has no sense. The authors didn’t explain in the text how they measured EDS spectra. They only mentioned the number of the spectra but no detail. For example, for the black glaze the variation of Pb is between 5.42 and 42.80 showing a very large one. This makes no sense. Unfortunately, I don’t recommend the paper to be published.
Author Response
Response to the Reviewer #1
We deeply appreciate the time the Reviewer spent reviewing our paper and the valuable recommendations he made. All the comments are taken into account and corresponding changes are made to the manuscript’s body text. For the convenience of the esteemed editor, we have divided all the reviewer's comments into 7 points. Detailed point-by-point answers are presented below. All the changes and additions to the text are colored in blue.
Comment #1.
Citation from the review: “…for the yellow color EDS analyses show the presence of Naples yellow which was not discussed in the text”.
Response to the comment #1:
This comment of the Reviewer #1 looks very interesting but needs some corrections and explanations.
1.The yellow color glaze was examined by pXRF, not by EDS (compare Tables 1 and 2). We could not apply SEM-EDS to the study of yellow colored area because of technical reason (complicated shape of sample unsuitable for putting into sampling box of SEM-EDS equipment).
- The section 4 Discussion contains the paragraph concerning to yellow glaze (lines 482-496).
- As known, the “Naples Yellow” is opaque glaze, or enamel, of Pb-Sb or Pb-Sb-Fe composition type. The presence of Sb is providing brightness of yellow color. We agree that image of yellow colored area of polychromic figurine looks to be similar to the “Naples Yellow” samples. However, the data on chemical composition of yellow glaze (Table 2) are not providing sure evidence for the interpretation of our case as true “Naples Yellow”.
- As known, the “Naples Yellow” glaze/enamel receipt was adopted firstly in China from European glazing technology in 17 century that is significantly later than the “red-and-green porcelain” appearance (end of 12 – the beg. of 13 c.). Our research object was found in a clearly stratified archaeological site, the time frame of which is determined by the 12–13 century. There is no reason to assume a later age of the artifacts from this site.
So, we can’t accept the idea about the interpretation of yellow glaze as “Naples Yellow”.
Comment #2.
Citation from review: “Black glaze contains iron and manganese, which is different from the red glaze’s composition”.
Response to the comment #2:
Thank the Reviewer for this comment!
The EDS examination did not detected Mn in elemental composition of black glaze and other glazes (see Table 1). The Mn was detected by pXRF in elemental composition of not only black glaze but of red, transparent, yellow and green glazes also. This chemical element is traditionally a concomitant element of iron in ores. Its presence in low concentrations does not cause doubts about its status as an "accompanying impurity". In all tested cases the content of Mn is insignificant, less than 1.0 % (Table 2). For instance, in black glaze the Mn content is 0.19–0.34 %. In general, in all cases the Mn is minor element not considered as probable coloring agent. Based on our knowledge and practical experience in the manufacture of pottery, we insist that manganese in such a low concentration cannot have any significant effect on the color characteristics of the glaze, its physical and chemical properties and technological features of the production behavior.
Comment #3.
Citation from review: “The content of Al and Si is important for the body analysis”.
Response to the comment #3:
The reviewer is right – the data on Al and Si content are important for the characteristics of body matrix (ceramics paste). The data on Al and Si in composition of body determined by EDS as well as data on Al : Si ratio are presented in sub-section 3.1. According to these data we consider body matrix as high-alumina clay with good refractory properties that is important for understanding of technological process of polychromic figurine making.
Comment #4.
Citation from review: ”…Slip layer contains a significant amount of zirconium which cannot be assumed as an impurity. An impurity refers to a concentration less than 1 %”.
Response to the comment #4:
In 3.2 (Slip) we note that Zr is detected in one only out of nine EDS spectra took for the slip layer. The spectrum with 13.86 % Zr is corresponding to the grain inclusion within SEM-EDS examined area.
As known, zirconium (mineral grains) occurs as the impurity in raw clays, and then zirconium inclusions may be detected in archaeological ceramics (Ref.20). It seems to be likely that we meet this case in our study.
We add more detailing comment in 3.2 paragraph + SEM photo with Zr inclusion (Fig.7).
Comment #5.
Citation from review:” For the glaze, alkali content (Na, K, and Mg) may be compared to the content of Pb, not Al and Si”
Response to the comment #5:
Thanks to the esteemed Reviewer for this valuable comment! We did not any comparing or correlation of Pb and Al-Si concentrations in the glazes. As for the alkali elements, the pXRF method because of some technical restrictions was not able to detect Mg, K, Na (comment in sub-section 2.2, lines 166-168). SEM-EDS determined only K in all examined cases (Table 1). The concentrations of this element are almost equal for the paste, slip, black and red glazes compositions. The Mg and Na were detected as the tenths or hundredths of percent. So, the information on the alkali elements seems to be not helpful for the consideration.
Comment #6.
Citation from review: “The authors didn’t explain in the text how they measured EDS spectra”……..
Thanks to the esteemed Reviewer for this comment!
Response to the comment #6:
The explanation concerning to the detail of EDS measurement is added to the manuscript (sub-section 2.2, lines 142-155).
Comment #7.
“…for the black glaze the variation of Pb is between 5.42 and 42.80 showing a very large one. …. This makes no sense.”
Response to the comment #7:
In our results we rely on the literature source [Ref. 7 - Jiang et al., 2017] studying the black glaze area for the "red-green porcelain" sample. In this investigation a large and similar range of values 5.05–36.78 % was noted for PbO in black glaze composition. Special consideration of black glaze composition case, in particular, Pb concentration, is presented in Section 4, lines 451-472.
Reviewer 2 Report
The object of this study is a Red-and-green porcelain figurine of Buddha from the Anan’evka walled settlement in the Pimor'ye region, which reflects the circulation of the products from the Central Plains to the Far East in the late Jin. However, there are several problems which need to be further considered and revised.
1. The EDS elemental composition data in Table 1 is not convincing. How many points are measured for the body, engobe, underglaze black and overglaze colors? How are they measured, from the cross-section or from the surface? Table 1 lists a fairly high content of Carbon, is this due to the carbon film plated on the sample for SEM/EDS testing? If so, you should remove the content of Carbon and renormalize the rest data.
2. The content of each major element in Table 1 gives a wide range with very large discreteness, which is beyond the acceptable level of normal testing. Therefore, it is necessary to re-examine the test data, and the current EDS data is not only not helpful to explain the chemical composition of the sample, but also misleading. If reliable EDS data cannot be guaranteed due to irregular sample shape or other reasons, I suggest that Table 1 be deleted.
3. For the PXRF data in Table 2, the test conditions need to be explained, for example, is it measured in air? What is the diameter of the beam spot? What reference samples are used for calibration?
4. Oxygen content is missing in Table 2. It is suggested that the results of element composition be converted into oxide composition according to the usual practice, and then normalized.
5. There is a problem with the explanation of the black color, the Pb in the black decoration data should come from the glaze above the black color rather than from the black color itself. Because the test points, such as points 4 and 5, are close to the yellow overglaze color, and the glaze surface is subjected to the Pb vapor condensation resulting from the yellow color, which is consistent with Reference [6].
Author Response
Response to the Reviewer #2
We deeply appreciate the time the Reviewer spent reviewing our paper and the valuable recommendations he made. All the comments are taken into account and corresponding changes are made to the manuscript’s body text. Detailed point-by-point answers are presented below. All the changes and additions to the text are colored in blue.
Comment #1.
Citation from review: “How many points are measured for the body, engobe, underglaze black and overglaze colors? How are they measured, from cross-section or from the surface?”
Response to the comment #1:
Thanks to the Reviewer for this certainly helpful comment! We add to the manuscript body text special methodical comments on sample preparing and examination (sub-section 2.2, lines 142-151 – for SEM-EDS examination, lines 173-176 – for pXRF examinations).
Comment #2.
Citation from review: “Table 1 lists a fairly high content of Carbon, …is this due to the carbon film plated on the sample…..?”
Response to the comment #2:
Thanks to the Reviewer for this helpful comment! No any special coatings/films (carbon, chromium, gold) were plated to examined sample before SEM-EDS analysis. We add special comment (Sub-section 2.2, Lines 146-147).
Comment #3.
Citation from review: “…each major element…gives a wide range with very large discreteness… If reliable EDS data cannot be guaranteed due to irregular sample shape (!) or other reasons I suggest that Table 1 be deleted.
Response to the comment #3:
Thanks to the Reviewer for this helpful comment! We accept this reviewer’s notion and the recommendation. We add some comment on the sample surface relief in the sub-section 2.1. (lines 91-92). We transform the data in Table 1 from the ranging to mean (average) values. Also, to increase the informativeness of the data and improve the interpretations we used additionally the method of visualizing Table 1 data by representing them in the format of "box-and-whiskers" plot diagram (lines 155-157; Fig.6, Fig.11).
Comment #4.
Citation from review: “…pXRF data…test conditions need to be explained… Calibration instrument?...”
Response to the comment #4:
Thanks to the Reviewer for this helpful comment! We have added a methodical description of pXRF calibration instrument and measurement procedure (sub-section 2.2, lines 164-166, 173-176).
Comment #5.
Citation from review: “It is suggested that the results of elemental composition be converted into oxide composition according to usual practice…”
Response to the comment #5:
Thanks to the Reviewer for this comment!
We agree with the Reviewer's comment that the data for the study of the composition of ceramic materials and glazes can be presented in the format of the oxide composition. However, in the present work the elemental composition in local areas was identified and it was obtained by SEM-EDS method. We consider that the presentation of the obtained data in the original format of chemical elements in this case is more correct and convenient.
Comment #6.
Citation from review: “…problem with the explanation of black color…”
Response to the comment #6:
Thanks very much to the Reviewer for his excellent and very valuable remark for our study! Following the comment, we added the interpretation of the obtained data (sub-section 3.3.2. – lines 310-315; section 4lines 451-472). Our examination data are well correlating with the results of X. Jiang et al., 2017 which the reviewer mentioned. This previous investigation results are really helpful as for the understanding of our research case.

Reviewer 3 Report
This paper applies a variety of analytical methods (SEM-EDS, PXRF, and optical microscopy) to fragments of a figurine from the Anan’evka walled settlement to determine the technology and composition of the clay and glazes on the object. The authors conclude that the technology (produced in a two part mold, with a first high temperature firing of the paste, black paint, and clear overglaze, followed by a low temperature firing after application of red, green, and yellow glazes) and form of the figurine (depicting the monk Budai) is consistent with production at contemporaneous Cizhou kilns in northern China as part of the “red-and-green porcelain/red-green ware” industry.
Much of the work presented here revolves around understanding the chemical constituents of different phases/components of the ceramics, for instance Fe in the black and red glazes, and Cu and Sb for the green and yellow components respectively. However, these measurements should be considered relatively qualitative given that the authors do not present any indication as to how accurate and precise their measurements are for the SEM or PXRF data. Was a quality control standard employed, and what instrument calibration was employed on the PXRF to generate data. How appropriate is this calibration for analyzing glassy materials? Without such information, it is hard to know whether to trust the actual numbers presented as representative of the actual composition of the different phases of the object. I also wonder if other techniques like portable FTIR or Raman might useful for understanding the mineralogical composition of the glazes to provide more insight into how they were produced. While ascription to the “red-and-green porcelain/red-green ware” tradition and the Cizhou kilns of northern China seems highly likely based on the available information (at least given that no other plausible alternative place of production is presented), is there any more specific elemental data that could be compared to from other studies to further demonstrate that this piece was actually made at the Cizhou kilns rather than representing an imitation? I found the description of this industry to be a bit scattered and hard to follow based on the text—in other words, was there any actual doubt as to where this object was manufactured, and does the presented evidence compellingly rule out any possible other production location?
Minor issues:
A figure showing the relative location of the Cizhou kilns (and any other possible alternative production locations) would be helpful.
A figure showing some of the evidence used to reconstruct the molding process would be helpful.
The paper needs some language editing in places as it is sometimes hard to follow.
Author Response
Response to the Reviewer #3
We deeply appreciate the time the Reviewer spent reviewing our paper and the valuable recommendations he made. All the comments are taken into account and corresponding changes are made to the manuscript’s body text. Detailed point-by-point answers are presented below. All the changes and additions to the text are colored in blue.
Comment #1.
Citation from review: “…authors do not present any indication as to how accurate and precise their measurements are for the SEM or pXRF data.”
Response to the comment #1:
Thanks to the Reviewer for this helpful comment!
We add methodical comments on the EDS and pXRF examination to the sub-section 2.2. Methods (lines 142-156, 157-176).
Comment #2.
Citation from review: “… what instrument calibration was employed on the pXRF to generate data?..”
Response to the comment #2:
Thanks to the Reviewer for this helpful comment! The portable analyzer was calibrated with the standard Olympus Analytical Instrument calibration check reference coin. Additions on calibration have been made to the text (sub-section 2.2, lines 164-166).
Comment #3.
Citation from review: “…I also wonder if other techniques like portable FTIR or Raman might useful for understanding the mineralogical composition of the glazes to provide more insight into how they were produced.… “
Response to the comment #3:
Thanks to the Reviewer for this helpful comment!
We agree absolutely with the notion about high effectiveness of Raman spectroscopy and FTIR method. We are familiar with the analytical capability of the approaches using these physicochemical methods. However, in presented study we used SEM-EDS and pXRF because of their current availability. Certainly, we plan to use Raman and FTIR methods in future.
Comment #4.
Citation from review: “…Is there any more specific elemental data that could be compared to from other studies to further demonstrate that this piece was actually made at the Cizhou kilns rather than representing an imitation?”
Response to the comment #4:
Thanks to the Reviewer for this really interesting comment!
This is interesting question taking into account that in ancient China the practice to make the imitations of famous ceramics or porcelains was known widely. However, as for the “red-and-green porcelain”, in the publications there are the data on its production only in the system of Cizhou kilns beginning from the 12-13 c. (for instance, the Ref. 4-9). No any other or alternative Chinese kilns centers, or sites, producing this kind of ceramics are known. We add some more information concerning to description of “red-and-green porcelain” production process in Cizhou kilns, please, see section 4, lines 425-430. Cizhou kilns system contained not only single kiln site Cizhou but the series of well-known kilns (Linshui, Guantai, and others) which produced certain kinds of ceramics, in particular, “red-and-green porcelain” according to common technological standards and receipts.
Comment #5.
Citation from review: “A figure showing the relative location of the Cizhou kilns (and any other possible alternative production locations) would be helpful”.
Response to the comment #5:
Thanks to the Reviewer for this helpful comment!
The map with Cizhou kilns area location is added to the manuscript (new Fig. 13, section 4). Because of no any alternative locations of “red-and-green porcelain” production are known they may be not marked at the map.
Comment #6.
Citation from review: “…A figure showing some of the evidence used to reconstruct the molding process would be helpful…”
Response to the comment #6:
Thanks to the Reviewer for this comment! Only evidence of molding process is presented by the impressions (footprints) of spatula-like tool ai the inner surface of the figurine body. The Fig.4-b is showing this evidence. This is single figure which could be presented as the illustration of molding process. In the text the references to molding technology in ancient Chinese ceramics production are in the Section 4, lines 377-379, 496.

Round 2
Reviewer 1 Report
I have some minor points to be improved. Then the paper may be published.
The scale bar in Figs. 2-4, is shown in mm. Is it correct? Can it be in cm? Please check it.
I didn't understand why the authors show only one data of Zr in comparison with the other elements which contain multiple data. This is not a representative result for the slip composition. Please revise the analysis of the slip layer in Table 1.
I think that in Table 2 the results of the elements are shown by their oxide forms (SiO2, Al2O3, PbO, etc.) because the oxide conversion factor is 2.13 for Si to SiO2. If this is in elementary form SiO2 should be around 150 % which makes no sense. Please revise the first row of Table 2.
Author Response
Response to the Reviewer #1
We deeply appreciate the time the Reviewer spent reviewing our paper and the valuable recommendations he made. All the comments are taken into account and detailed answers are presented below.
Comment #1.
The scale bar in Figs. 2-4, is shown in mm. Is it correct? Can it be in cm? Please check it.
Response to the comment #1:
Thanks to the Reviewer for his comment! We agree with the esteemed Reviewer that the version we used on the scale bar may be confusing to some readers. We have changed the designs of the scale bars in Figures 2–4 to make them clear.
Comment #2.
I didn't understand why the authors show only one data of Zr in comparison with the other elements which contain multiple data. This is not a representative result for the slip composition. Please revise the analysis of the slip layer in Table 1.
Response to the comment #2:
Thank the Reviewer for his comment!
The Zr in the slip material is found in single EDS spectra. It indicates the nature of the Zr distribution in the material bulk – the distribution in the form of separately encountered inclusions, which is confirmed by the SEM image on the figure 8, that shows these inclusions occur as small zirconium microparticles < 10 μm. Following the Reviewer comment the Table 1 data are corrected – the Zr column showing not mean value is deleted. The Fig 6b concerning the slip composition is corrected also.
Comment #3.
I think that in Table 2 the results of the elements are shown by their oxide forms (SiO2, Al2O3, PbO, etc.) because the oxide conversion factor is 2.13 for Si to SiO2. If this is in elementary form SiO2 should be around 150 % which makes no sense. Please revise the first row of Table 2.
Response to the comment #3:
We thank the esteemed Reviewer for his comment!
Table 2 shows the results of elemental composition (not oxide composition!) obtained by semi-quantitative pXRF analysis. We leave these data in elemental composition format to show the entire spectrum of the elements that have been identified. In the scientific literature on the study of ceramic materials, there is a presentation of the component composition both as oxides [DOI: 10.1140/epjp/s13360-022-02669-9] and as elements [DOI: 10.1016/j.ceramint.2018.09.213]. This is normal practice for the study of ceramic-type materials in the case of portable devices of this method. In our works, we prefer to use the chemical composition representation in the form of an elemental composition.

Reviewer 2 Report
Reasonable answers have been given to the questions raised, and the quality of the manuscript has been significantly improved.
Author Response
Thanks to the esteemed reviewer for his time and attention!
We improved the English quality of our manuscript by the kind assistance of a native speaker, Dr. Richard L. Bland (University of Oregon, Eugene, OR, USA). The English language is highlighted in yellow.
Reviewer 3 Report
The authors appear to have addressed the issues that I raised during review of the initial submission by providing more information on instrument settings, including box-and-whisker plots which make the results easier to follow and understand, and providing more background information on comparative analyses of similar ceramics.
Author Response

(The authors gave the same response as above.)
